# Functional Characterization of Novel *MC4R* Variants Identified in Two Unrelated Patients with Morbid Obesity in Qatar

**DOI:** 10.3390/ijms242216361

**Published:** 2023-11-15

**Authors:** Idris Mohammed, Senthil Selvaraj, Wesam S. Ahmed, Tara Al-Barazenji, Ayat S Hammad, Hajar Dauleh, Luis R. Saraiva, Mashael Al-Shafai, Khalid Hussain

**Affiliations:** 1College of Health & Life Sciences, Hamad Bin Khalifa University, Doha P.O. Box 34110, Qatar; imohammed-c@sidra.org (I.M.); wisamyejo@yahoo.com (W.S.A.); saraivalmr@gmail.com (L.R.S.); 2Division of Endocrinology, Department of Pediatric Medicine, Sidra Medicine, Doha P.O. Box 26999, Qatar; hdauleh@sidra.org; 3Department of Disease Modeling and Therapeutics, Sidra Medicine, Doha P.O. Box 26999, Qatar; sselvaraj@sidra.org; 4Department of Biomedical Sciences, College of Health Sciences, QU Health, Qatar University, Doha P.O. Box 2713, Qatar; ta1706200@student.qu.edu.qa (T.A.-B.); ayat.hammad@qu.edu.qa (A.S.H.); 5Biomedical Research Center, Qatar University, Doha P.O. Box 2713, Qatar

**Keywords:** *MC4R*, monogenic obesity, severe obesity, childhood obesity, Qatar

## Abstract

The leptin–melanocortin pathway is pivotal in appetite and energy homeostasis. Pathogenic variants in genes involved in this pathway lead to severe early-onset monogenic obesity (MO). The *MC4R* gene plays a central role in leptin–melanocortin signaling, and heterozygous variants in this gene are the most common cause of MO. A targeted gene panel consisting of 52 obesity-related genes was used to screen for variants associated with obesity. Variants were analyzed and filtered to identify potential disease-causing activity and validated using Sanger sequencing. We identified two novel heterozygous variants, c.253A>G p.Ser85Gly and c.802T>C p.Tyr268His, in the *MC4R* gene in two unrelated patients with morbid obesity and evaluated the functional impact of these variants. The impact of the variants on the *MC4R* gene was assessed using in silico prediction tools and molecular dynamics simulation. To further study the pathogenicity of the identified variants, GT1-7 cells were transfected with plasmid DNA encoding either wild-type or mutant *MC4R* variants. The effects of allelic variations in the *MC4R* gene on cAMP synthesis, *MC4R* protein level, and activation of PKA, ERB, and CREB signaling pathways in both stimulated and unstimulated ɑ-MSH paradigms were determined for their functional implications. In silico analysis suggested that the variants destabilized the *MC4R* structure and affected the overall dynamics of the *MC4R* protein, possibly leading to intracellular receptor retention. In vitro analysis of the functional impact of these variants showed a significant reduction in cell surface receptor expression and impaired extracellular ligand binding activity, leading to reduced cAMP production. Our analysis shows that the variants do not affect total protein expression; however, they are predicted to affect the post-translational localization of the *MC4R* protein to the cell surface and impair downstream signaling cascades such as PKA, ERK, and CREB signaling pathways. This finding might help our patients to benefit from the novel therapeutic advances for monogenic forms of obesity.

## 1. Introduction

Obesity is a complex condition caused by genetic, lifestyle, and environmental factors, which has become a significant health problem worldwide [1]. Monogenic obesity due to single-gene pathogenic variants in the leptin–melanocortin pathway, an essential energy homeostasis pathway, accounts for 6% of the total cases of severe early-onset obesity [2]. The melanocortin-4 receptor (*MC4R*) gene is a crucial component in this pathway and is predominantly expressed in the hypothalamus’s paraventricular nuclei (PVN). The first evidence supporting the association of *MC4R* with obesity was seen in mice in 1997 [3] and subsequently in humans in 1998 by two independent groups [4,5]. These studies showed that target disruption of the *MC4R* or frameshift mutation in the gene causes severe obesity accompanied by hyperphagia in mice and humans [3,4,5]. The *MC4R* gene is a member of the G-protein-coupled receptor (GPCR) that binds to its endogenous ligand, the alpha-melanocyte stimulating hormone (α-MSH) and activates adenylate cyclase-3 (*ADCY3*). The activation of *ADCY3* subsequently converts adenosine triphosphate (ATP) into cyclic monophosphate (cAMP), a secondary messenger vital in various downstream signaling cascades [6]. The augmented intracellular cAMP levels, in turn, stimulate the activation of the downstream effectors such as protein kinase A (PKA), extracellular-signal-regulated kinase (ERK) and cyclic-AMP response element binding protein (CREB), which play an integral role in regulating energy intake and expenditure, thus contributing to the pathogenesis of obesity [7,8].

The *MC4R* gene, a one-exon gene, is localized on the long arm of human chromosome 18, 18q21.3. The gene encodes a 332-amino-acid transmembrane receptor protein. The *MC4R* receptor binds to α-MSH, a peptide produced due to the cleavage of the *POMC* precursor by the pro-hormone convertase PC1/3 and PC2 [9]. The *MC4R* gene regulates energy homeostasis by decreasing appetite and increasing satiety signaling downstream of the *POMC* neurons [10]. Pathogenic variants with complete or partial loss of function in the *MC4R* gene are the most common cause of monogenic obesity, accounting for up to 6% of severe early-onset obesity due to monogenic obesity genes [11]. These genetic variations lead to reduced protein expression, hindered α-MSH binding, altered receptor trafficking, or inefficient coupling with the stimulatory G-protein, Gαs [12,13]. Patients carrying disease-causing variants in the *MC4R* gene are characterized by extreme obesity, hyperphagia, and increased linear growth [14].

In most cases, the mode of inheritance is autosomal dominant, while less frequently, patients exhibit an autosomal recessive inheritance pattern [15]. Patients with heterozygous *MC4R* variants are more common than homozygous or compound heterozygous carriers, where the latter develop a more severe form of obesity [3,16]. So far, numerous heterozygous mutations in *MC4R* linked to obesity with variable severity have been studied functionally using in vitro cell models, revealing a subset that disrupts G-protein-coupled receptor (GPCR) signaling [17,18].

In this study, we performed clinical, genetic, and biochemical investigations of two patients in Qatar who exhibited pronounced early-onset obesity due to novel variants in the *MC4R* gene. We conducted in vitro functional characterization, in silico prediction tools, and molecular dynamic simulations to determine the pathogenicity and contribution of the variants to severe obesity. The graphic overview of the research design is illustrated in Figure 1.

## 2. Results and Discussion

### 2.1. Case Presentation

Case 1: Patient 1 is an 11-year-old Jordanian girl with morbid obesity. The patient had a birth weight of 3.5 kg and started to gain weight when she was a few months old. Her current weight is 72.5 kg, her BMI is 36.4 Kg/m^2^ (Z-score of 2.88), and she has a BMI percentile of 99.8th centile. She has a strong family history of severe obesity; her mother and father (first cousin) had gastric bypass surgery for morbid obesity. The patient has two siblings who are also obese. All her baseline investigations were normal.

Case 2: Patient 2 is a 17-year-old Qatari boy with morbid obesity. His birth weight was 4 Kg, and he was born to consanguineous parents. He started to gain weight at the age of 2 years; his current weight is 277 Kg with a BMI of 88.4 Kg/m^2^ (Z-score of +3.8) and a BMI percentile >99.99th centile. He has marked hyperphagia, difficulty breathing, and elevated liver enzymes (ALT: 85 U/L (5–30 U/L), AST: 128 U/L (0–39 U/L), ALP: 191 U/L (52–171 U/L). His mother, who is overweight, does not carry the variant, and the phenotype and DNA of the father were not available for assessment. Table 1 summarizes the biochemical investigations of the two cases. Figure 2 shows the pedigree of the two cases.

### 2.2. Genetic Analysis

A targeted gene panel sequencing of 52 genes associated with obesity revealed two novel missense variants in *MC4R*, c.253A>G p.Ser85Gly, and c. 802T>C p.Tyr268H. Patient 1 had a compound heterozygous p.Ser85Gly and p.Tyr268. His variants were inherited from both parents, who remained obese despite undergoing bariatric surgery (gastric bypass). Patient 2 had a heterozygous missense variant p.Tyr268His. The mother did not carry the variant, and the father’s DNA was unavailable for genetic analysis. A search in public databases such as gnomAD V2.1.1, 1000Genomes, TOPMED, and GME Variome verified that these variants were not previously reported in the literature. To assess the pathogenic mechanism of these two *MC4R* variants, we performed in silico analysis; the variant p.Ser85Gly was predicted to affect the polarity of the protein due to the replacement of a highly polar (serine) to a nonpolar (glycine) residue. The variant is located on the second transmembrane helix of the *MC4R* gene in a highly conserved residue among diverse species, suggesting it could have a deleterious effect on the protein. The second novel variant, p.Tyr268His, resulted in amino acid substituting tyrosine (neutral) with histidine (positively charged) on residue 268 on the sixth transmembrane helix. These novel variants were strongly predicted to be deleterious to the *MC4R* gene using three independent in silico prediction tools: SIFT, Polyphyne-2, and MutationTaster (Table 2).

### 2.3. Prediction Analysis and Molecular Dynamics Simulation

We used online structure-based prediction tools to investigate the effect of the identified mutations on the structural stability of the *MC4R* protein [19,20,21,22,23,24]. Results showed a destabilizing impact of these mutations on the protein structure (Table 3). However, these tools do not provide a detailed picture of the structural dynamics of the complex. To obtain a clear picture of how these mutations affect the structural dynamics, we performed 100 ns all-atom, explicit-solvent MD simulations of the WT and mutant complexes (Figure 3). Analyzing MD trajectories showed an overall increase in the structural dynamics of the *MC4R* mutants, as indicated by the increased RMSD and DCC values of the protein. No similar changes were observed in the peptide ligand (Figure 3 and Figure 4). These destabilization effects could potentially impact the localization of the *MC4R* in the cell membrane and/or hinder the signal transmission that follows ligand binding.

### 2.4. The S85G and Y268H Mutants Reduce MC4R Cell Surface Expression

GT1-7 cells derived from the murine hypothalamus are a widely recognized model for examining melanocortin-associated signaling pathways and energy homeostasis mechanisms. Thus, we used GT1-7 and overexpressed either the wild-type or mutant *MC4R* or empty vector (control). First, we sought to investigate the impact of S85G and Y268H substitution on the total protein level of *Mc4R* in cells with or without stimulating α-MSH. Notably, the level of *MC4R* was not altered in GT1-7 cells transfected with mutant *MC4R* compared with WT-transfected cells, suggesting that S85G and Y268H substitution did not affect protein expression (Figure 5A,B). Moreover, no significant difference was observed in the total *MC4R* level in the cells after stimulation with α-MSH, indicating that ligand stimulation did not alter the *MC4R* protein synthesis.

Our next step was to delve into the membrane localization of the *MC4R* receptor. To this end, cell surface protein biotinylation was used to isolate the membrane *MC4R* in wild-type or mutant *MC4R* transfected cells with or without α-MSH stimulation. Interestingly, the membrane localization of *MC4R* was significantly increased in cells transfected with WT-*MC4R* compared with the control. In contrast, S85G and Y268H substitution significantly affected the membrane localization of mutant *MC4R* when compared with control and WT-*MC4R*, suggesting that the mutant *MC4R* proteins are likely to be retained intracellularly, hinting at potential challenges in achieving proper membrane localization (Figure 5C,D). Moreover, no significant difference was observed in *MC4R* membrane localization in cells stimulated with α-MSH when compared with their respective untreated group.

### 2.5. The S85G and Y268H Mutants Decrease Agonist-Stimulated cAMP Accumulation

To understand the effect of S85G and Y268H variants on *MC4R* functionality, we transfected GT1-7 cells with either the wild-type *MC4R* (WT-*MC4R*) or its mutant counterparts. Post transfection, cells were treated with 0.5 mM IBMX for 10 min and subsequently incubated with 100 nM α-MSH for an additional 15 min. The 3-isobutyl-1-methylxanthine (IBMX) serves as a nonspecific phosphodiesterase (PDE) inhibitor, working to halt the breakdown of cyclic adenosine monophosphate (cAMP)—a vital messenger molecule within cells. By inhibiting PDE activity, IBMX causes an uptick in the cellular cAMP concentration, amplifying the cellular response to hormonal signaling. On the other hand, Alpha-MSH (α-MSH) operates as a natural ligand for the *MC4R* receptor. Upon binding, it activates *MC4R*, which triggers an elevation in intracellular cAMP levels. Utilizing both IBMX and α-MSH offers a dual advantage: IBMX amplifies the cell’s general sensitivity to hormonal signals, while α-MSH specifically targets and stimulates the *MC4R*. This combined approach is frequently employed to ensure potent *MC4R* stimulation and to heighten the cAMP levels observable in assays. While the basal cAMP levels remained unaltered, a distinct response pattern emerged upon agonist stimulation. Cells overexpressing the WT-*MC4R* displayed a pronounced surge in cAMP production. In contrast, cells bearing the novel *MC4R* variants did not exhibit a significant increase in cAMP levels when compared with their respective unstimulated controls (Figure 6).

### 2.6. Mutant MC4R Fails to Activate the Downstream Signaling

To investigate the downstream signaling dynamics mediated by mutant *MC4R*, GT1-7 cells were transfected with either WT-*MC4R* or its mutant variants. A total of 36–48 h following transfection, these cells were exposed to 100 nm α-MSH for 3 h. Consistent with the cAMP data, α-MSH stimulation led to a marked elevation in PKA and ERK1/2 phosphorylation in cells expressing WT-*MC4R*, as opposed to their unstimulated counterparts. Intriguingly, this agonist-mediated activation was conspicuously absent in cells harboring the mutant *MC4R* (Figure 7A–D). Similarly, when observing CREB phosphorylation, cells transfected with WT-*MC4R* exhibited a significant boost in response to α-MSH stimulation. In contrast, cells expressing the mutant *MC4R*s demonstrated minimal reactivity (Figure 7E,F). This pattern indicates that the mutations could potentially hinder *MC4R*’s capacity to relay downstream signaling in the presence of the agonist effectively. The mechanism by which loss-of-function variants on the *MC4R* gene inactivate the downstream signaling cascade and dysregulate energy balance is summarized in Figure 8.

## 3. Discussion

Monogenic obesity due to *MC4R* pathogenic variants is the most common cause of severe childhood obesity [2]. We identified two novel missense variants in two unrelated patients (a Qatari and a Jordanian) with severe early-onset obesity. The patients described in this report resembled previously described cases of *MC4R* patients with early-onset obesity, hyperphagia, and increased linear growth [25]. Patient 1 carried a compound heterozygous mutation (S85G and Y268H), while patient 2 had a heterozygous Y268H variant. To date, there are more than 200 *MC4R* variants reported in the literature, out of which the homozygous and compound heterozygous variants are very rare but responsible for more severe phenotypes compared to heterozygous variants [26]. The mechanism by which the vast majority of these variants exert their effect is believed to result from haploinsufficiency and loss of gene function. Conversely, the gain of function variants in the *MC4R* gene were found to be protective against obesity and negatively associated with the phenotype [25].

Generally, conserved amino acids are anticipated to affect protein function and stability significantly. The amino acids found in our patients, serine at position 85 and tyrosine at position 268 in the *MC4R* gene, are highly conserved residues in the melanocortin receptors among different species [27]. In addition, these two variants are located on the transmembrane helices of the *MC4R* gene. Transmembrane helices are crucial in the membrane proteins’ structure and folding. Many variants that are localized on transmembrane helices are known to affect the localization of the protein and hinder the cell membrane signaling, consequently leading to severe obesity [2,28,29].

GPCR receptors, when binding to their ligand, activate a cascade of downstream signaling; one of the most common and vital genes in these cascades is ADCY3, which converts ATP into a secondary messenger, cAMP [30]. The upregulation of cAMP plays a crucial role in various downstream signaling mechanisms, notably protein kinase A (PKA) [31]. Once activated, PKA can then phosphorylate a diverse array of target proteins, ultimately leading to changes in cellular activity, including transcription, metabolism, and ion channel activities [32]. Conversely, pathogenic variants on the *MC4R* gene have been shown to have a profound impact on cAMP production, subsequently impairing the downstream of the cAMP-PKA signaling mechanism that affects energy expenditure [32,33,34]. Both variants identified in our study showed decreased cAMP generation even after stimulation with a-MSH. This finding is in concordance with previously reported cases that showed *MC4R* variants that lead to reduced cAMP generation cause severe obesity [8,29,35].

The agonist stimulation of *MC4R* has multifaceted downstream effects, including the activation of PKA, ERK, and CREB, which play pivotal roles in translating *MC4R* signaling into cellular responses. When *MC4R* is stimulated by its agonists, it activates a cascade of intracellular events that eventually lead to PKA activation and subsequently ERK1/2 activation [30]. The activated ERK can translocate to the nucleus, where it can phosphorylate and activate various transcription factors, leading to the expression of target genes associated with cellular growth, differentiation, and survival [36]. Parallel to the ERK pathway, *MC4R* activation also leads to the phosphorylation and activation of the cAMP response element-binding protein (CREB). The activation of CREB in the context of *MC4R* signaling is particularly relevant as it connects the dots between *MC4R* activation, and the regulation of genes associated with energy balance and appetite [32]. In vitro studies of the mutant *MC4R* variants demonstrated the minimal reactivity of p-PKA, p-ERK1/2, and p-CREB compared to the wild-type *MC4R*, which exhibited a significant boost in response to α-MSH stimulation. These findings suggest that the mutations could potentially hinder *MC4R*’s capacity to activate downstream signaling in the presence of the agonist effectively.

Functional in vitro studies of the two novel variants, p.Ser85Gly and p.Tyr268His, showed a loss of signal transduction activity of the mutant receptors. Despite the total protein expression of the wild-type and mutant alleles being comparably similar, the mutant variants’ cell surface protein expression showed reduced expression compared to the wild-type cells upon stimulation with α-MSH, suggesting that the variants impair ligand binding. To date, most of the *MC4R* pathogenic variants that lead to loss of function were found to be affecting ligand binding affinity to the membrane receptor or the intracellular retention of the receptor due to misfolding and trafficking [37]. Receptor misfolding and intracellular retention of the *MC4R* gene are the most common mechanisms that lead to severe childhood obesity [18]. Thus, we assume the reduction in the cell membrane protein expression in the mutant cells in our study could probably be due to the retention of the misfolded protein in the endoplasmic reticulum and proteasomal degradation.

We performed a molecular simulation dynamics prediction of the variants on *MC4R* protein structure. As shown previously, the transmembrane helix contains a remarkably high number of conserved residues compared to N-terminal and C-terminal loops [38]. We tested the impact of the identified mutations on the structural stability of the *MC4R* protein, and the result, in accordance with previous reports, showed that the variants would not only destabilize the protein structure but also increase the overall structure dynamics of the *MC4R* [39].

Interestingly, variable expressivity and incomplete penetrance have been reported in patients with mutations in the GPCRs, which could be due to mutations in modifier genes [10]. Even though both of our patients carry a shared variant, p.Y268P, the severity in patient 2 (17 years old) is very remarkable (BMI 88.4 Kg/m^2^) compared with patient 1 (11-years old, BMI 36.4 Kg/m^2^), this could potentially be explained by the fact that the severity of the phenotype might be exacerbated with age or due to variable expressivity of the variant.

Some in vitro studies showed that chemical chaperones and a newly FDA-approved drug, Setmelanotide (an *MC4R* agonist), could be used to treat some monogenic forms of obesity due to POMC, PCSK1, or LEPR deficiency. Setmelanotide can rescue some *MC4R* proteins retained in the endoplasmic reticulum and traffic the receptor to the plasma membrane via cell surface relocalization [6], whereas chemical chaperones are chemicals that resemble endoplasmic reticulum chaperones which have the ability to reach the central nervous system. These chemicals are selective and work for a subset of patients who have obesity due to *MC4R* receptor deficiencies. Molecular chaperones promote the folding of the *MC4R*, stabilize misfolded *MC4R*s, and rescue misfolded *MC4R* receptors of the cell membrane [40,41]. Further studies are required to understand whether obese subjects carrying variants like those identified in our patients affecting the membrane expression of *MC4R* may benefit from clinical chaperones or Setmelanotide for weight loss.

## 4. Materials and Methods

This study was approved by the Institutional Review Board (IRB) for the protection of human subjects in Sidra Medicine, Qatar (IRB reference number 1689931). Written consent was obtained from the patients and their parents for their participation in this study. Peripheral blood was collected from patients and available family members in EDTA tubes for DNA extraction. According to the manufacturer’s recommendation, genomic DNA was extracted from peripheral blood using the QIAamp DNA blood midi kit (Cat. 51185, Qiagen, Hilden, Germany). The concentrations and purities were assessed using a Nanodrop 2000 spectrophotometer (ThermoFisher Scientific, Waltham, MA, USA). For Next-generation sequencing, exonic regions of all genes of interest were captured using an optimized set of DNA hybridization probes. The captured DNA was sequenced via massively parallel sequencing using the Illumina NovaSeq 6000 platform reversible dye terminator (RDT) (Illumina, San Diego, CA, USA); detailed DNA sequencing and sequencing analysis methodologies are described in our recently published article [42].

### 4.1. MC4R Cloning

The *MC4R* wild-type and mutant cDNA (full length) were cloned into pcDNA3.1-C-(k)DYK vectors. These constructs were purchased from (GenScript, Singapore). The plasmid contains DYKDDDK epitope, AMPr (ampicillin-resistant) for bacterial selection, and NeoR (Neomicin-resistant) for mammalian cell selection. The constructs were cloned in a CMV-driven and T7 promotor with a sequence of (TAATACGACTCACTATAG). The vector contained six restriction enzyme sites for Nhe I, Afl II, Hind III, Kpn I, and BamH I. The target gene was cloned after the Kozak sequence (GCCACC) in an open reading frame and was tagged with the DYK tag (DYKDDDDK). The DNA plasmids were transformed into chemically competent *E. coli* (Sigma-Aldrich, Saint Louis, MO, USA).

### 4.2. Cell Culture and Transfection

GT1-7 mouse hypothalamic GnRH neuronal cells, obtained from Sigma-Aldrich, were cultured in Dulbecco’s Modified Eagle’s Medium (DMEM)-high glucose, enriched with 10% fetal bovine serum (FBS), 2 mM L-Glutamine and 1% penicillin/streptomycin. The cells were kept at 37 °C in a humidified atmosphere containing 5% CO_2_. The growth medium was replenished with fresh medium every 48 h while subculturing activities were carried out upon reaching 80% confluency. Transient transfection was carried out using lipofectamine 3000 according to the manufacturer’s instructions, and 36 h post transfection, the cells were used for the downstream analysis.

### 4.3. Isolation of Total/Cell Surface Proteins and Western Blotting

After transfecting GT1-7 cells with either wild-type or mutant *MC4R*, we extracted total protein by lysing the cells with RIPA buffer. For isolating cell surface proteins, we used the Pierce Cell Surface Protein Isolation Kit (ThermoFisher Scientific, Waltham, MA, USA) in accordance with the manufacturer’s instructions. First, the cells were chilled at 4 °C for 15–20 min and then biotinylated with EZ-Link Sulfo-NHS-SS-Biotin for 30 min at the same temperature. We then stopped the biotinylation process with a quenching buffer and lysed the cells on ice using the kit-provided lysis buffer. The biotinylated proteins were isolated by incubating the lysates with NeutrAvidin Agarose, washed thrice, and eluted using the kit’s elution buffer, heated at 95 °C for 5 min. The total and cell surface proteins were loaded onto a 4–12% Nupage gel and subsequently transferred onto a Polyvinylidene Fluoride (PVDF) membrane, blocked with a non-fat milk buffer, and incubated overnight at 4 °C with the appropriate primary antibody. The membrane was then incubated with an HRP-conjugated secondary antibody, and proteins were visualized using Chemdoc (Bio-Rad Laboratories, Hercules, CA, USA) and quantified using ImageJ (NIH, Bethesda, MD, USA).

### 4.4. cAMP Assay

The cAMP assay was performed using the cAMP ELISA Kit (Colorimetric) from (Cell Biolabs Inc., San Diego, CA, USA) according to the manufacturer’s instructions. GT1-7 cells were cultured in a 96-well plate until 80–90% confluence and then transfected with wild-type or mutant *MC4R* constructs. Cells were exposed to 0.5 mM IBMX for 10 min, followed by subsequent stimulation with 100 nM α-MSH for 15 min. Post stimulation, cells were lysed, and lysates were added to the pre-coated cAMP ELISA plate. The cAMP present in the lysate competed with cAMP conjugate for the anti-cAMP antibody binding sites during an incubation period at room temperature. After washing away unbound components, we added an HRP-conjugated secondary antibody and incubated the plate again for 2 h. Following incubation, we added a colorimetric substrate for HRP, and the reaction was suspended after an appropriate period. The intensity of the color developed was inversely proportional to the amount of cAMP in the sample. The plate was read at 450 nm, and the cAMP concentration in samples was determined by comparing with the standards.

### 4.5. Structure-Based Prediction of Mutations on Protein Stability

Since the crystal structure of *MC4R* in complex with its native ligand, alpha-melanocyte stimulating hormone (α-MSH), is not available, we instead used the recently resolved crystal structure of *MC4R* in complex with a synthetic analog of α-MSH (Nle4, D-Phe7)-α-MSH (i.e., NDP-α-MSH), to study the structural and ligand-binding changes induced by the mutations in the complex [17]. The crystal structure of the *MC4R*-NDP-α-MSH complex was obtained from the RCSB database (PDB: 7PIV, Chain P: NDP-α-MSH (aa 1–13), Chain R: *MC4R* (aa 40–316) and Ca^2+^ (co-factor)). The effect of the mutations on the stability of the complex was assessed using computational prediction tools such as mCSM [19], Maestro [21], I-Mutant2.0 [22], and MUpro [24].

### 4.6. Molecular Dynamics Simulation

All-atom, explicit solvent, molecular dynamics simulations of *MC4R* in complex with NDP-α-MSH were performed using NAMD 3.0 software [43] and CHARMM36m force field [44]. The topology and parameter input files required to simulate the wild-type (WT) and mutant complexes were generated using the CHARMM-GUI server [45]. Briefly, the complex structure was solvated in a TIP3P cubic water box [46] with at least 10 Å distance between any of the atoms in the complex and the edge of the water box. The biomolecular simulation system was then subjected to energy minimization and thermal equilibration with periodic boundary conditions [47]. This was followed by 100 ns production simulation runs. The integration timestep was set up at 2 fs. A 12 Å cut off with a 10 Å switching distance was chosen to handle short-range non-bonded interactions, while long-range non-bonded electrostatic interactions were conducted using a particle-mesh scheme at 1 Å PME grid spacing [48,49,50,51]. MD trajectories were analyzed using the available tools in VMD. Dynamic cross-correlation (DCC) analysis was performed for *MC4R* Cα atoms using the Bio3D R package (version 3.1.0). Results were represented as heatmaps that indicate the range of correlations from −1 to +1. A cut off of 0.8 was applied to the positive and negative DCC values to investigate the changes in the lower and upper DCC extremes, and the results were represented as graphic images.

### 4.7. Statical Analysis

Data represent means ± SEM of three independent experiments performed in duplicate. Asterisks indicate a significant difference using one-way ANOVA followed by Dunnett’s analysis.

## 5. Conclusions

In summary, we identified two novel *MC4R* variants in patients with severe obesity. Both in silico investigation and in vitro functional assessment substantiate the pathogenicity of the identified variants. While these variants do not affect total protein levels, they notably reduce the cell surface expression of the mutant proteins. The decreased cell surface expression is likely attributable to the post-translational modification causing the misfolding of the mutant *MC4R* protein and subsequent retention within the endoplasmic reticulum. The identification of these variants in the population of Qatar further stresses the importance of looking into the genetic cause of obesity in patients presenting with severe early-onset obesity. These insights may pave the way for our patients to capitalize on the cutting-edge therapeutic advances tailored for monogenic obesity, such as clinical chaperones and setmelanotide.

## Figures and Tables

**Figure 1 ijms-24-16361-f001:**
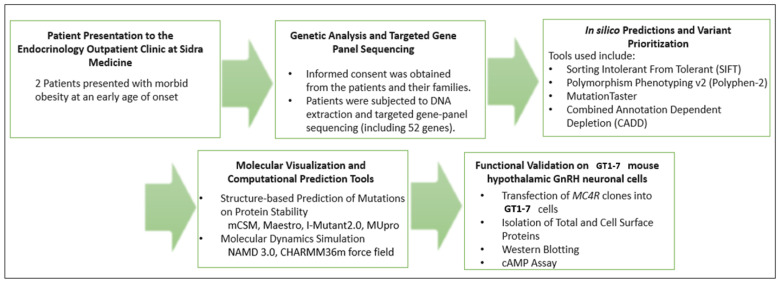
Flowchart illustration of the research design of the study.

**Figure 2 ijms-24-16361-f002:**
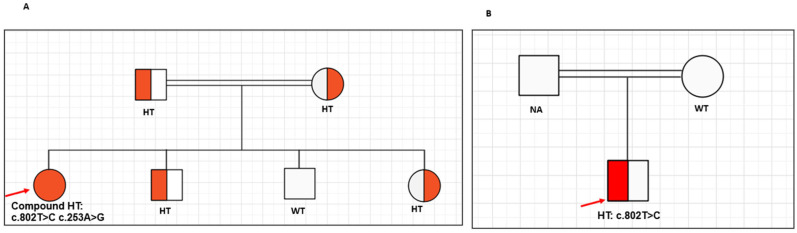
Pedigree of the two patients. (**A**) shows the pedigree for compound heterozygous patient (both mother and father are heterozygous, and two of her siblings are also heterozygous. Mother, father, and one of the heterozygous siblings has undergone bariatric surgery). (**B**) shows the pedigree for the heterozygous patient, the mother does not carry the variant and the DNA for the father was not available for analysis. Note: NA: not available, HT: heterozygous, WT: wild type; red arrow indicates the proband.

**Figure 3 ijms-24-16361-f003:**
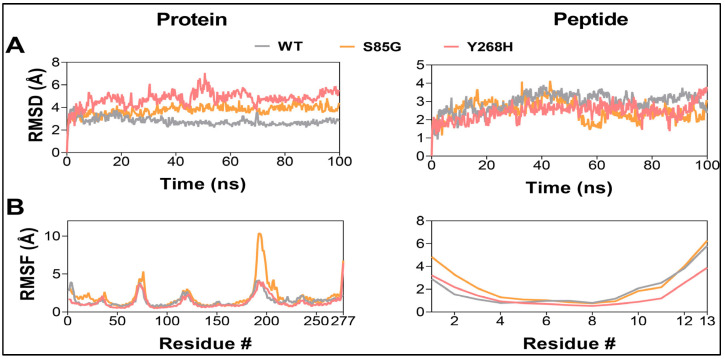
*MC4R* mutants result in a destabilization of the *MC4R* protein. Root-mean-square deviation analysis (**A**) and Root-mean-square fluctuation values (**B**) of *MC4R* (**left**) and peptide ligand (**right**) in the WT (gray), S85G (yellow), and Y268H (red) protein–peptide complexes.

**Figure 4 ijms-24-16361-f004:**
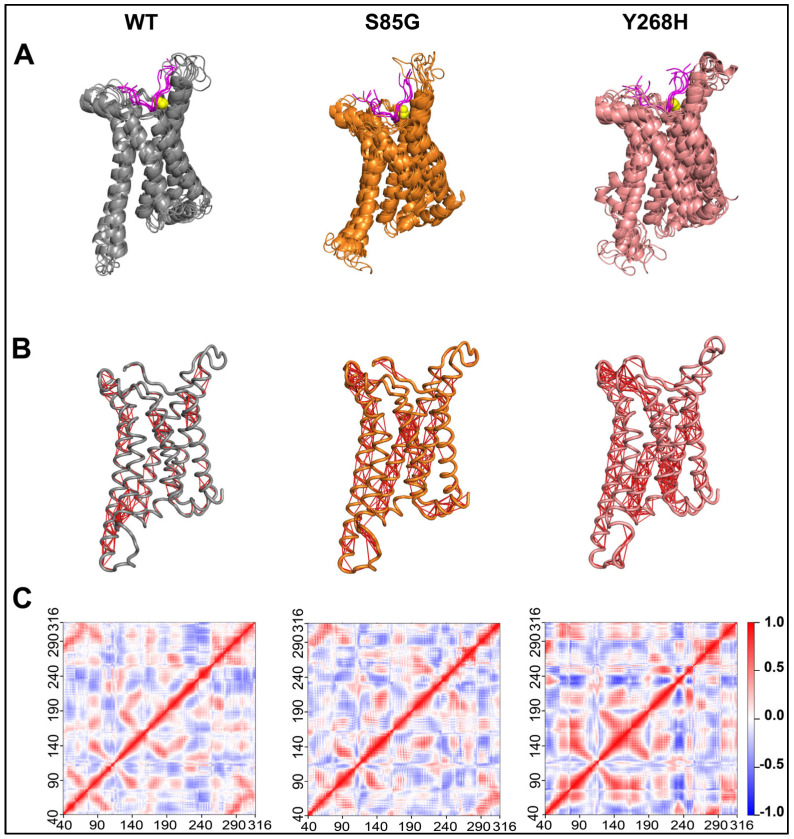
The *MC4R* mutations increase the structural dynamics of the protein. (**A**) Composite image of graphic snapshots representing 100 ns of MD simulation, taken 20 ns apart. (**B**) Graphic image representing *MC4R* Cα atom dynamic cross-correlation (DCC) values obtained from 100 ns MD simulations after applying a DCC value cut off of ±0.8. (**C**) Heatmap showing *MC4R* Cα atom DCC values obtained from 100 ns MD simulations of the *MC4R* WT (**left panel**), *MC4R*(S85G) (**middle panel**), and Y268H (**right panel**). Note the impact of the mutations on the overall dynamics of the protein.

**Figure 5 ijms-24-16361-f005:**
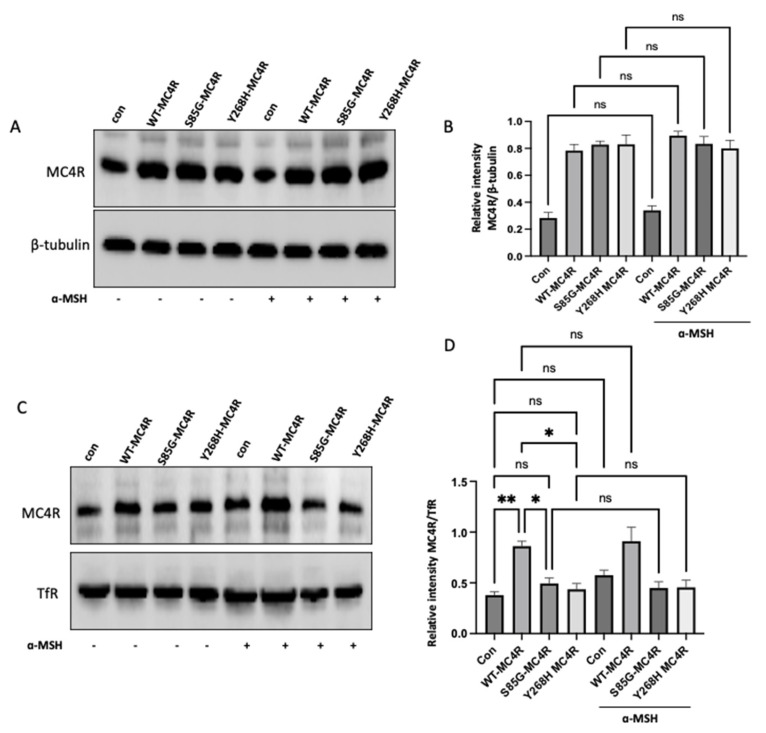
Analysis of *MC4R* expression and membrane localization in GT1-7 cells. (**A**) Western blot analysis showing the total protein levels of *MC4R* in GT1-7 cells transfected with wild-type (WT) *MC4R*, mutant *MC4R* (S85G and Y268H), or empty vector (control) and then stimulated with 100 nm α-MSH for 12 h after 36–48 h. Representative immunoblots shown were probed for *MC4R* and β-tubulin (loading control marker for the whole cell lysates). (**B**) Quantitative analysis of the *MC4R* protein expression derived from independent experiments. The densitometric evaluation was used to ascertain the relative intensity of *MC4R* bands against the β-tubulin signals, serving as the normalization factor. Graphically illustrated data represent mean ± SEM gathered from at least three independent experiments. (**C**) GT1-7 cells were transfected with either WT or mutant *MC4R*. After 36–48 h of transfection, cells were stimulated with 100 nm α-MSH for 12 h. Representative Western blot images illustrating the membrane localization of *MC4R* following cell surface protein biotinylation. TfR (loading control marker for membrane fraction). (**D**) Densitometric quantification of membrane-bound *MC4R* based on several independent experiments. The intensity of *MC4R* bands was normalized to the signals from TfR. The compiled data, representing the mean ± SEM, was sourced from at least three individual experiments. Asterisks indicate a significant difference using one-way ANOVA followed by Dunnett’s analysis (* *p* ≤ 0.05, ** *p* ≤ 0.01). Note: ns: not significant.

**Figure 6 ijms-24-16361-f006:**
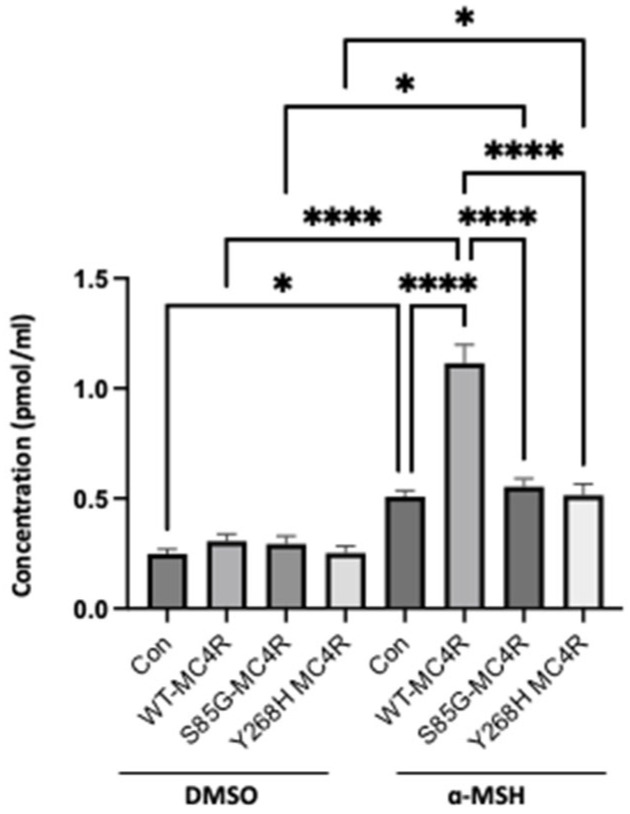
*MC4R* variants diminish cAMP accumulation. GT1-7 cells were transfected with either WT-*MC4R* or mutant *MC4R* variants and subsequently subjected to IBMX treatment (0.5 mM) for 10 min, followed by a 15 min incubation with 100 nM α-MSH. Intracellular cAMP content was measured by ELISA. Data represent means ± SEM of three independent experiments performed in duplicate. Asterisks indicate a significant difference using one-way ANOVA followed by Dunnett’s analysis (* *p* ≤ 0.05, **** *p* ≤ 0.0001).

**Figure 7 ijms-24-16361-f007:**
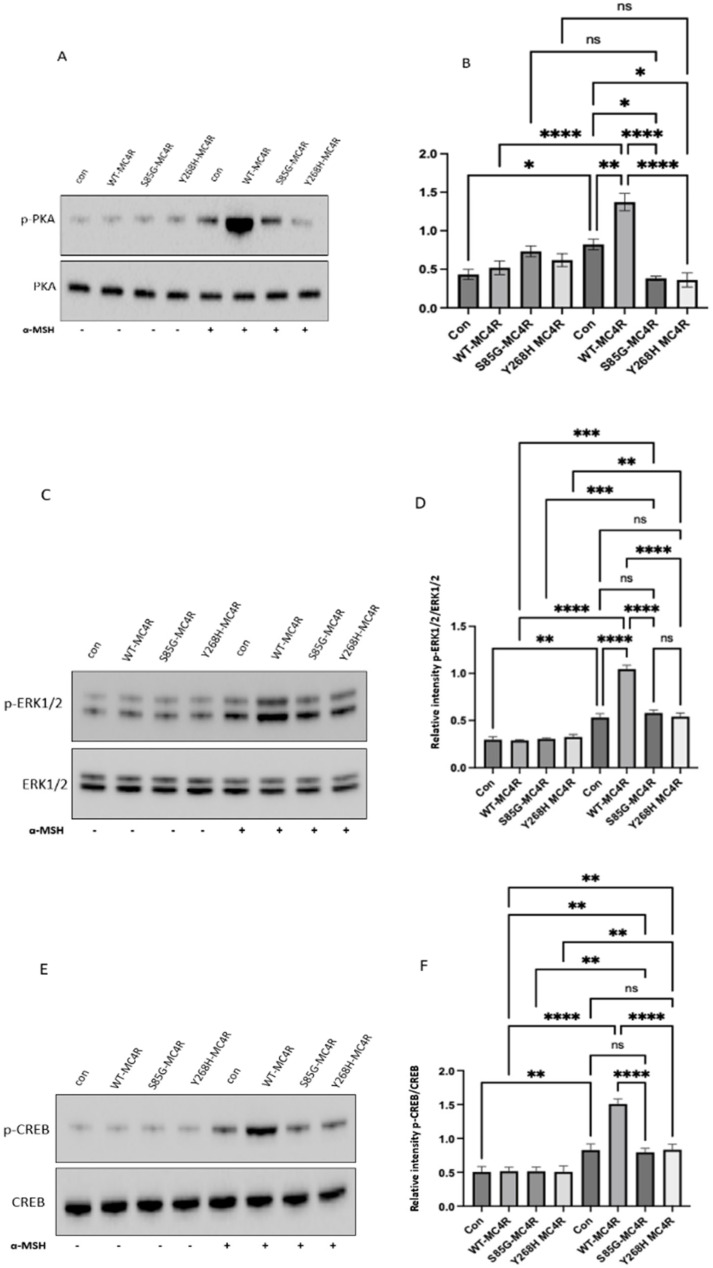
*MC4R* variants impair the PKA, ERK1/2, and CREB activation in GT1-7 cells. (**A**,**C**) Representative Western blot images showing the levels of phosphorylated PKA (Thr197), total PKA, phosphorylated ERK1/2 (pThr202/Tyr204-ERK1/2), and total ERK in GT1-7 cells transfected with either wild-type *MC4R* (WT-*MC4R*) or its mutant counterparts. Following a 36–48 h post-transfection period, cells were exposed to 100 nm α-MSH for 3 h. (**E**) Analogous Western blot images illustrating the levels of phosphorylated CREB (pSer133-CREB) and total CREB post transfection and α-MSH treatment. (**B**,**D**,**F**) Densitometric analyses of p-PKA, p-ERK1/2, and p-CREB signals normalized to their respective total protein levels. Data represent means ± SEM of three independent experiments. Asterisks indicate a significant difference using one-way ANOVA followed by Dunnett’s analysis (* *p* ≤ 0.05, ** *p* ≤ 0.01, *** *p* ≤ 0.001 **** *p* ≤ 0.0001).

**Figure 8 ijms-24-16361-f008:**
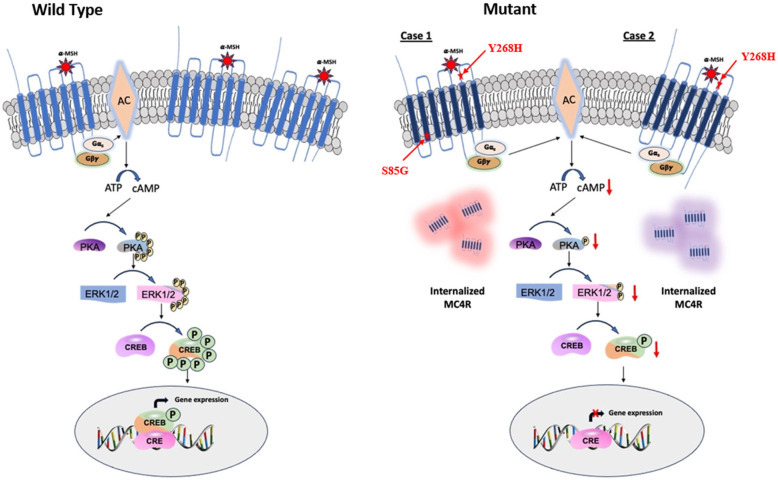
Schematic representation of agonist-induced *MC4R* signaling in energy homeostasis. *MC4R* mutant-decreased intracellular cAMP leads to downregulation of PKA, which, in turn, inactivates ERK1/2 and CREB and imbalances the energy homeostasis.

**Table 1 ijms-24-16361-t001:** Clinical and biochemical features of the two cases.

Test	Value			
	Patient 1	Reference	Patient 2	Reference
Age of onset	3 months		2 years	
BMI (Kg/m^2^)	36.4		88.4	
ALT	14	10–25 U/L	85 U/L (H)	5–30 U/L
AST	20	20–38 U/L	128 U/L (H)	0–39 U/L
GGT	15	6–18 U/L	NA	NA
ALP	NA	NA	191 U/L (H)	52–171 U/L
HBA1c	5.70	<6.0%	5.40	<6.0%
Total cholesterol	5	3.1–5.9 mmol/L	5.11	3.1–5.9 mmol/L
Trig	1.8	0.6–2.5 mmol/L	2.5 (H)	1.8–2.2 mmol/L
HDL	1.1	0.9–1.7 mmol/L	0.3 (L)	0.9–1.7 mmol/L
LDL	3.6	1.4–4.2 mmol/L	3.7 (H)	<3.4 mmol/L
TSH	2.80	0.76–4.64 mIU/L	4.58 (H)	0.5–4.3 mIU/L
Free T4	12.7	8.1–14.9 pmol/L	13.1	12.9–20.6 pmol/L
Insulin	NA	NA	41.1	1.4–47 mc unit/mL
Leptin	NA	NA	34 (H)	0.7–5.3 ng/mL

Note: NA: not available.

**Table 2 ijms-24-16361-t002:** Summary of the in silico predictions of the two novel *MC4R* variants.

Chromosomal Location (GRCh37)	*MC4R* Variant	Amino Acid Change	SIFT	Polyphen-2	Mutation Taster	gnomAD MAF	GME MAF
Chr18: 58039330	c.253A>G	p.Ser85Gly	Deleterious	Probably Damaging	Disease-Causing	0	0
Chr18:58038781	c.802T>C	p.Thy268His	Deleterious	Probably Damaging	Disease-Causing	0	0

Note: GME Variome: The Greater Middle East, MAF: Minor Allele frequency.

**Table 3 ijms-24-16361-t003:** Prediction of the changes in Gibb’s free energy (kcal/mol) induced by the indicated single point mutation obtained using various bioinformatics prediction tools.

Bioinformatics Tool	*MC4R* (S85G)	Outcome	*MC4R* (Y268H)	Outcome
ΔΔG mCSM	−1.282	Destabilizing	−1.81	Destabilizing
ΔΔG MUpro	−1.54	Destabilizing	−1.28	Destabilizing
ΔΔG I-Mutant 2.0	−2.33	Destabilizing	−1.83	Destabilizing
ΔΔG Maestro	1.9994	Destabilizing	2.85	Destabilizing

## Data Availability

All relevant data are included in the manuscript. Further original data will be made available by contacting the corresponding authors within the regulations of the ethical approval.

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
