# Peer review of "Functional Characterization of Novel MC4R Variants Identified in Two Unrelated Patients with Morbid Obesity in Qatar"

_ijms, 2023, doi:10.3390/ijms242216361_

Round 1

Reviewer 1 Report

Comments and Suggestions for Authors

In this study, the authors identified two novel MC4R variants in patients with severe obesity. Both in silico investigation and in-vitro functional assessment substantiate the pathogenicity of the variants identified. While these variants do not affect total protein levels, they notably reduce the cell surface expression of the mutant proteins. The conclusion shows great clinic implications. In addition, their data look convincing. However, this manuscript is lack of organization, such as the introduction (see below). Also, PKA assessment experiment show be provided regarding to the working model.

P1L23-24: The authors say they found two novel MC4R variants and then followed by the sentence “A targeted gene 23 panel consisting of 52 obesity-related genes was undertaken“. It is a little bit confused. Please rephrase or use connectors or linking words.

P1L35-36: ”In-vitro analysis assays and in silico analysis revealed the pathogenicity of the two novel MC4R variants identified”. This is redundant words. Please rephrase or delete.

P1L44-46: “leptin-melanocortin pathway…accounts for 6% of severe early-onset obesity cases (2)”
P2L68-69: “… MC4R…accounting for up to 6% of the severe early-onset form of obesity (11)”
Do authors mean that the single-gene pathogenic variants in the leptin-melanocortin pathway are all MC4R pathogenic variants?

P2L53: “The MC4R gene is a member of the G-protein coupled receptor (GPCR)”
P2L77-79: “ …numerous heterozygous mutations in MC4R ……revealing a subset that disrupts G-protein-coupled receptor (GPCRs) signaling.”
I’m confused by authors saying “…revealing a subset that disrupts GPCRs signaling” without any references.

P2L79-82:”However, approximately 25% of … such as endocytosis and disruption of receptor homodimerization (17)”. Please delete or move to discussion, since authors didn’t talk about this scenario in their own study.

P4L180-183: Please indicate the difference of Asterisk number through all the figures with Statical Analysis.

P4L194: “his current weight is 277 with BMI of 88.4 Kg/m2”. Please add the unit “277 kg”.

P5L284: Please give some information of the agonist “IBMX” used in the assay.

Figure3: Please mention that what are the β-tubulin and TfR marked ,like that β-tubulin and TfR marked were blotted as markers for cytoplasmic and membrane fractions”

Figure6: The working model shows MC4R mutants decreased intracellular cAMP leads to downregulation of PKA. However, this study doesn’t show the data. Please add.

Reviewer 2 Report

Comments and Suggestions for Authors

This study discovered two new mutation sites in the MC4R gene, a major obesity candidate gene, through sequencing of obesity candidate genes in two severely obese patients, and examined their effects on protein structure, intracellular localization, and cell signaling effects.

My overall impression of the paper is that it combines very detailed experimental method techniques and the latest technology at each stage, allowing a sufficient understanding of the cell biological effects of mutations.

However, I have some questions on the results that require explanation as follows.

1. The criteria for selecting the 52 obesity candidate genes and their references do not seem to be clear.

2. It is stated that the parents of pediatric patients with severe obesity were both severely obese. The genetic characteristics of the parents were briefly described in the first paragraph of the results, but there is a need to express them more easily through a family tree.

3. In this study, two mutations in MC4R were selected, and although they may affect intracellular signaling mechanisms, both are understood to be heterozygote. Then, I wonder whether the wild type allele among heterozygotes can compensate for the insufficient function of the mutant allele.

4. This study conducts a wide variety of analyzes from sequencing to cell experiments, so the result description steps are very confusing while reading the paper. It would be much easier to understand if there were study design pictures of the entire experiment process.

Round 2

Reviewer 2 Report

Comments and Suggestions for Authors

I think the authors have answered my questions sufficiently.